# Arterial Digital Pulse Photoplethysmography in Patients with Suspected Thoracic Outlet Syndrome: A Study of the “Ca+Pra” Maneuver

**DOI:** 10.3390/diagnostics11061128

**Published:** 2021-06-21

**Authors:** Jeanne Hersant, Pierre Ramondou, Francine Thouveny, Mickael Daligault, Mathieu Feuilloy, Patrick Saulnier, Pierre Abraham, Samir Henni

**Affiliations:** 1Vascular Medicine Department, University Hospital in Angers, 49100 Angers, France; Jeanne.hersant@chu-angers.fr (J.H.); Pierre.ramondou@chu-angers.fr (P.R.); Samir.Henni@chu-angers.fr (S.H.); 2University Angers, UMR CNRS 6015-INSERM 1083, MITOVASC, SFR ICAT, 49045 Angers, France; mickael.daligault@chu-angers.fr; 3Radiology Department, University Hospital in Angers, 49100 Angers, France; Francine.thouveny@chu-angers.fr; 4Thoracic and Vascular Surgery Department, University Hospital in Angers, 49100 Angers, France; 5School of Electronics (ESEO), Universite catholique de l’ouest, 49100 Angers, France; Mathieu.feuilloy@eseo.fr; 6University Le Mans, LAUM CNR S6613, 72000 Le Mans, France; 7University Angers, Inserm, 1066 CNRS 6021, MINT, SFR ICAT, F-49045 Angers, France; patrick.saulnier@univ-angers.fr; 8Biostatistics Department, University Hospital in Angers, 49100 Angers, France; 9Sports and Exercise Medicine Department, University Hospital in Angers, 49100 Angers, France

**Keywords:** thoracic outlet syndrome, transcutaneous oximetry, photoplethysmography, pathophysiology, ischemia, arterial inflow

## Abstract

The level of pulse amplitude (PA) change in arterial digital pulse plethysmography (A-PPG) that should be used to diagnose thoracic outlet syndrome (TOS) is debated. We hypothesized that a modification of the Roos test (by moving the arms forward, mimicking a prayer position (“Pra”)) releasing an eventual compression that occurs in the surrender/candlestick position (“Ca”) would facilitate interpretation of A-PPG results. In 52 subjects, we determined the optimal PA change from rest to predict compression at imaging (ultrasonography +/− angiography) with receiver operating characteristics (ROC). “Pra”-PA was set as 100%, and PA was expressed in normalized amplitude (NA) units. Imaging found arterial compression in 23 upper limbs. The area under ROC was 0.765 ± 0.065 (*p* < 0.0001), resulting in a 91.4% sensitivity and a 60.9% specificity for an increase of fewer than 3 NA from rest during “Ca”, while results were 17.4% and 98.8%, respectively, for the 75% PA decrease previously proposed in the literature. A-PPG during a “Ca+Pra” test provides demonstrable proof of inflow impairment and increases the sensitivity of A-PPG for the detection of arterial compression as determined by imaging. The absence of an increase in PA during the “Ca” phase of the “Ca+Pra” maneuver should be considered indicative of arterial inflow impairment.

## 1. Introduction

Thoracic outlet syndrome (TOS) is difficult to diagnose [1,2,3,4,5]. Among the various attitudinal stress tests used in patients with suspected TOS, the Roos test, originally described in the 1960s, also called EAST (“elevated arm stress test” or “extended arm stress test”) or the 90° EAR (abduction external rotation) test, is largely used. In patients with suspected TOS, ultrasonography (US) enables the manual recording of arterial inflow during the Roos test [6,7]. US allows for the detection of the level of compression and the presence of eventual complications (aneurysms, thrombosis), but it is a hand-hold technique, requires trained operators, and can only be performed on one side at a time. Arterial digital pulse plethysmography (A-PPG) allows recordable and observer-independent measurements and has been proposed in TOS diagnosis [8,9]. Nevertheless, no consensus has been reached on the normal limit in defining inflow impairment [8,9,10,11,12]. Gergoudis and Barnes defined an abnormal A-PPG response as a ≥75% reduction in digital pulse amplitude [13]; Geven et al. defined the loss of pulsatility as a remaining amplitude <5% of the resting value [9], while Adam considered any dampened or reduced waveform as abnormal [14].

Arm elevation induces a physiological increase in A-PPG pulse amplitude (PA) in normal subjects [9,15,16]. Therefore, it would be expected that any decrease or even an impaired increase in A-PPG during the Roos test performed in the sitting or standing position is an abnormal response in patients with suspected TOS. Nevertheless, to date it has been impossible to determine whether the increase in PA has reached its maximum while arms are elevated in the surrender/candlestick (“Ca”) position or whether it is blunted by a certain degree of arterial compression. In brief, we believe that the major reason why no consensus exists about what should be considered an abnormal A-PPG response during the Roos test is because: first, A-PPG is a semi-quantitative tool, and second, the individual response resulting from arm elevation but without a 90° abduction and external rotation is unknown.

We hypothesized that, after the “Ca” position, keeping arms elevated but with elbows and hands in front of the patient as in a prayer position (“Pra”) would reveal normal inflow during arm elevation with A-PPG. Indeed, if PA increases from “Ca” to “Pra”, this suggests that the PA increase was not optimal in “Ca” because arterial compression impaired inflow. To test these hypotheses, we performed a prospective interventional study recording PA with A-PPG at the finger level in patients with suspected TOS and in a group of apparently asymptomatic healthy volunteers. Our aim was to define the optimal cut-off point to be used for A-PPG in detecting an arterial compression during the “Ca+Pra” maneuver.

## 2. Materials and Methods

### 2.1. Experimental Design

We recruited 31 patients that were referred to our laboratory for the investigation of symptoms suggestive of the presence of TOS, and 21 normal asymptomatic (apparently healthy) subjects. After oral and written explanation of the protocol, individual written informed consent was required for inclusion. We recorded age, sex, weight, height, and systolic and diastolic arm pressure on both sides. Patients self-completed the “disability of the arm and shoulder” 38-item questionnaire (DASH). The DASH score was calculated if at least 90% of the answers to the first 30 questions were available. Ultrasound results (and potential arterial radiological imaging in patients) were retrieved from the patient’s file, and ultrasound imaging was performed per the protocol in healthy subjects. All healthy subjects had an ultrasound investigation performed by an independent physician before the PPG recording. Results were recorded as positive or negative for the presence of arterial unilateral or bilateral compression during positional maneuvers arm by arm on either ultrasound imaging or arteriography. Patients unable to understand the information for linguistic or cognitive reasons, as well as patients under 18 years of age, were not included in the analysis. The protocol was performed blinded to the results of the investigations performed during the routine visit to the patients.

### 2.2. Attitudinal Maneuvers

The Roos test is largely used in the evaluation of patients with suspected TOS. It has been previously shown that arm elevation results in a physiological increase in pulse amplitude (PA) in normal subjects [15,16]. Therefore, we performed a candlestick-prayer (“Ca+Pra”) maneuver: a modified version of the Roos test during which the surrender/candlestick position used for the Roos tests (“Ca”) is maintained for 30 s (without opening and closing of hands to avoid movement artifacts on A-PPG recordings). After 30 s, there was a change to the prayer position (“Pra”), without lowering the hands and with elbows in front of the patient, which was maintained for 15 s. In the “Pra” position, the elbow and hands are at the same level relative to the heart level as for the “Ca” position. The purpose of the “Pra” position is to open the costoclavicular angle and attain arm elevation without vascular compression. Its specific goal in the present study was to confirm whether pulse amplitude in the “Ca” position was, or was not, normally increased. If not, pulse amplitude would remain unchanged between the “Ca” and “Pra” positions, while it would increase during the “Pra” position if amplification with arm elevation was moderately impaired during the “Ca” position, as presented in Figure 1. After 45 s, the upper limbs were lowered.

### 2.3. Photoplethysmography Recordings

We performed the arterial pulse photoplethysmography (A-PPG) on the second finger of both hands using adult finger soft-tip SpO2 sensors (Sino-K, Shenzhen, CN) on a 50 Hz basis. The recording was started at least 30 s before the start of the provocative maneuver and stopped at least one minute after the end of the provocative maneuver. Each recording enables the detection of the A-PPG signal during each cardiac cycle. Arterial pulse amplitude (PA) was determined as the difference between the maximal and minimal A-PPG value over all 1.5 s intervals.

Moving averaging on pulse amplitude was applied over each series of 10 consecutive points. A transient artifact peak was systematically observed at upper limb elevation and lowering. Then, PA at rest and PA during the “Ca” position were calculated between 20 and 5 s before, and between 5 and 20 s after, the maneuver was started to avoid the movement artifact influencing recorded results. Since arm elevation results in a physiological increase in PA in normal subjects [15,16], we normalized PA as a percentage of the PA observed during the “Pra” position (between 35 and 40 s) after the beginning of the provocative maneuver. For the comparison with DROP results, PA changes (PAc) were expressed as the difference between PA during the “Ca” position and PA observed at rest, and were expressed in normalized amplitude (NA) units.

### 2.4. Statistical Analysis

Kolmogorov–Smirnov tests were used to test the distribution of variables, and results are presented as mean +/− standard deviation (SD) for parametric, or median (25°/75° centiles) for non-parametric, continuous variables. Analysis of variance (ANOVA) was used to compare the effect of the “Ca-Pra” maneuver on A-PPG in the healthy volunteers and in the patients with suspected TOS. We used the receiver operating characteristics (ROC) and area under curve (AUC) technique to determine the performance of A-PPG in detecting the presence or absence of a positive imaging (ultrasound or angiography). Thereafter, we calculated the sensitivity, specificity, and negative and positive predictive values, as well as the accuracy (with a binomial exact calculation of a 95% confidence interval) of each criterion in predicting the presence of a compression on ultrasound or radiological imaging. Lastly, we compared the performance of our results to those obtained using the normal limits proposed by Gergoudis (≥75% reduction in PA) or by Geven (>95% reduction of PPG amplitude). All statistical analyses were performed using SPSS (IBM SPSS statistics V15.0, Chicago, IL, USA) and the EasyROC online calculator (http://www.biosoft.hacettepe.edu.tr/ accessed on 1 March 2021) [17]. A comparison of the ROC curve was performed according to the method proposed by Hanley and McNeil [18]. We estimated that 20% of arterial compression would be observed and expected the AUC for A-PPG to be at least 0.700. A minimal number of 90 observations was required for a type I error of 5% and 80% power. For all tests, a two-tailed *p* < 0.05 was used to indicate statistical significance.

## 3. Results

Between March 2018 and December 2020, we recruited 52 subjects. We studied 31 different patients (13 males, 18 females). Patients were 41.3 ± 11.9 years old, with weights of 71.5 ± 12.7 kg and heights of 169 ± 9 cm. Systolic and diastolic pressures were 131 ± 14 mmHg and 80 ± 9 mmHg on the right side and 131 ± 17 mmHg and 84 ± 12 mmHg on the left side, with no patient showing a difference in arm pressure of more than 20 mmHg. All but two of the patients were right-handed. The DASH score of the subjects was 28 ± 21% with four missing scores. Nine were off work, six of whom because of their upper limb pain. Eleven patients had right unilateral, five left unilateral, and six bilateral pain or discomfort. Ten took pain killers on a regular basis because of pain or discomfort. Positional tests during their medical routine visit reproduced usual symptoms in all but five of the patients. In 15 of the patients with suspected TOS, ultrasound or angiographic investigations confirmed the presence of arterial compression on one or both sides (20 arms), as shown in Figure 2.

The 21 young healthy subjects (10 males, 11 females) were 27.0 ± 2.9 years old, with weights of 63.3 ± 10.6 kg and heights of 172 ± 9 cm. Systolic and diastolic pressures were 114 ± 8 mmHg and 75 ± 7 mmHg on the right side and 112 ± 10 mmHg and 73 ± 7 mmHg on the left side, with no patient showing a difference in arm pressure of more than 20 mmHg. All healthy subjects were asymptomatic by history, but one had a unilateral and one had a bilateral positive ultrasound result, and those two patients developed symptoms of forearm fatigue and discomfort during the attitudinal tests.

A typical example of a patient with unilateral right arterial compression during arm elevation is shown in Figure 3.

As shown in Figure 4, absolute values were slightly, but not significantly, lower in subjects with suspected TOS than in the healthy subjects both at rest and during the candlestick maneuver. Nevertheless, no difference was observed between the two groups in values found at the “Pra” phase of the test.

Imaging (ultrasound or angiography) showed the presence of a compression in 20 of the 62 upper limbs studied in patients with suspected TOS but in only 3 of the 42 upper limbs in the healthy subjects. Therefore, the prevalence of positive imaging was 22.1% among the 104 studied upper limbs. Using the results of ultrasound and angiography to characterize the presence of a compression with ROC analysis, the area under the ROC curve observed for the different ways of expressing the results are shown in Table 1.

As shown in Figure 5, the optimal cut-off point determined from the ROC curve was an absolute increase of fewer than 3 NA from rest.

Using this cut-off point resulted in a 91.4% (83.0/96.5) sensitivity, 60.9% (38.5/80.3) specificity, 89.2% (76.8/95.5) positive predictive value, and 66.7% (48.0/84.0) negative predictive value in predicting a positive result at imaging. As a comparison, the sensitivity, specificity, positive predictive value, negative predictive value, and accuracy were: 0.0% (0.0/14.5), 98.8% (93.2/100.0), 0.0% (0.0/97.5), and 77.7% (68.4/85.3) for the criteria proposed by Geven, and were 17.4% (5.0/38.8), 98.8% (93.3/100.0), 80.0% (28.4/99.5), and 80.8% (73.1/88.6) for the criteria proposed by Gergoudis, respectively.

## 4. Discussion

Diagnosing TOS is difficult [19], and many instances of TOS include intricate signs of arterial and neurogenic compression [4,20], which explains the significant interest in improving arterial investigations in patients with suspected TOS [20,21]. The routine clinical approaches to attitudinal arterial compression are manual pulse palpation or sub-clavicular auscultation during the dynamic maneuvers. These are obviously simple, of low cost and accurate, but not recordable. An electronic stethoscope could allow continuous recordings of the sub-clavicular bruit related to sub-clavicular compression but, to the best of our knowledge, this has never been proposed. Ultrasound allows recordings and remains relatively simple [22,23]. Nevertheless, it is a manual technique, and it can only measure one side at a time. Arterial pulse amplitude (PA) estimation by digital A-PPG allows bilateral recordings and is an attractive tool in the context of TOS [9]. Previous authors have used the A-PPG technique that allows objective and simultaneous recording of pulse throughout the provocative test [9,12]. Raising the arm above heart level increased the systolic amplitude of the finger pulse amplitude with A-PPG by 56 to 70% in normal subjects, but with wide variability [16,24]. These changes likely result from changes in transmural pressure because they disappear when transmural pressure is maintained constant [25]. On the contrary, attitudinal sub-occlusion or occlusion of the sub-clavicular artery is expected to result in decreased finger A-PPG amplitude [9,13]. Consequently, normal and occlusive results have an opposite effect on the A-PPG signal. Nevertheless, at the individual level, because A-PPG is a semi-quantitative technique and lacking the expected normal PA, it remains unclear to what extent a PA change of the A-PPG signal is indicative of an abnormal response.

We believe that the “Pra” position is an innovative way of quantifying the expected normal PA amplification and of defining whether the PA change observed during the “Ca” position used for the Roos test was normal or not. Contrary to previous authors [9,13], we observed that even a moderate increase in PA (when expressed in NA values) can be indicative of an abnormal hemodynamic response, with PA in the “Ca” position being much lower than in the “Pra” position in these cases. Indeed, in the absence of arterial compression, PA should remain unchanged between the “Ca” and the “Pra” positions, since the level of the hand compared to heart level is unchanged. Therefore, an increase in the PA observed between the “Ca” and the “Pra” positions likely reflects a blunted response in “Ca” resulting from an arterial compression. There is particular interest in the “Pra” position after the candlestick position and the present study confirm that an impaired increase (< 3 NA, that corresponds almost to the absence of a PA increase from rest) in PA is of diagnostic interest. This corroborates the results proposed by Adam et al. [14], because we have provided objective proof of the expected amplification at the individual level. Obviously, the criterion proposed by Geven et al. [9] was inappropriate when applied to our population, with the difference between maximal and minimal PPG signal over 1.5 s never being less than 5% of the difference observed at rest. We believe that this is a result of a relatively high signal to noise ratio and could be of technical origin. It is important to note that the “Ca+Pra” maneuver requires the patient to be standing or sitting, while previous use of A-PPG was sometimes performed in the lying position [12]. The sensitivity of A-PPG was four times higher with our analysis than that with the criterion proposed by Gergoudis. This was apparently associated with an important decrease in the positive predictive value. Nevertheless, the number of positive results with the Gergoudis criterion is very small and the precision in the determination of the positive predictive value is low (95% CI: 28.4/99.5%). Furthermore, from a clinical perspective, we believe that it is important to select those patients that would require invasive imaging better and that sensitivity is therefore the most important index. Due to the small number of subjects, the ROC curve obtained from normalized values, although providing the larger AUC, did not reach a significant difference from the area resulting from the other methods (absolute changes or percent changes in non-normalized values). Analysis from a larger population is required.

Another point of interest is that values tended to be lower on average at rest and during the candlestick position in patients with suspected TOS compared to the healthy subjects. It is possible that this difference results from the difference in age between the two groups. Another explanation could be due to a chronic compression of the neurovascular bundle when the arms are in the resting position in patients with suspected TOS. We doubt that this is the explanation because no difference was found between arms with positive imaging and normal arms. Nevertheless, this hypothesis remains to be tested, and an increase in PA in patients with suspected TOS during a Cyriax release test [26] could provide evidence for this hypothesis.

There are limitations to the present work.

Firstly, we report and compare our results to angiography in only some of the patients with suspected TOS. Indeed, radiological imaging is lacking in most of our patients. Surgery in our experience is only proposed after active rehabilitation has failed to improve the symptoms and was unsuccessful to deal with the complaints of the patients. Angiography is only used as a pre-surgical test, and many patients with TOS do not undergo surgery. Finally, the question remains whether or not a comparison of the clinical tests performed in the standing position can be compared properly to radiological images that are performed in patients that are lying down, since it was previously shown that arteriography in the supine position underestimates the prevalence and severity of attitudinal compression compared to images performed in the sitting position [27].

Secondly, many other tests (Adson, Eden, Allen, Wright tests, etc.) have been proposed. Whether or not A-PPG recordings during other maneuvers would provide similar results remains to be investigated.

Thirdly, compression of the neural or vascular structures may occur at various levels as they pass through narrow passageways to the arm, including the inter-scalene triangle, the costoclavicular space, the sub-coracoid space (pectoralis minor compression), or in front of the humeral head. Compression may also result from the presence of abnormal bands, tumors, cervical ribs, and bone prominence following healing of fractures. These different entities may not be affected the same way by the prayer position. Future studies are required to analyze the response to the Ca+Pra maneuver in perspective of the level(s) and nature of arterial compression(s).

Fourthly, symptoms were not systematically associated with the presence of inflow impairment in our group of patients with suspected TOS, underlying the importance of the holistic approach in the diagnosis of TOS and the need for objective proof of the presence of ischemia in defining the arterial origin of TOS [19].

Finally, it could be suggested that other techniques, such as strain gauge plethysmography and laser Doppler, might provide more quantitative or more reliable results. This is possible, but A-PPG, is a low-cost and readily accessible tool. Nevertheless, technical improvements are required to reduce the A-PPG artifacts induced by the movements when changing from one position to another.

## 5. Conclusions

From a physiological perspective, the “Ca+Pra” maneuver appears to be an interesting modification of the Roos test (elevated arm stress test), and of major interest for A-PPG recordings. It enables an improved quantitative analysis of A-PPG results, facilitates the interpretation of A-PPG results in TOS, and improves the sensitivity of A-PPG in predicting positive imaging. In practice, attitudinal arterial compression during TOS can result in either incomplete (stenosis) or complete (occlusion) compression of the sub-clavicular artery, and then in different degrees of inflow impairment.

This ability to quantify inflow impairment with A-PPG during the “Ca+Pra” maneuver might also be of critical clinical interest in comparing the level of inflow impairment with A-PPG to the degree of ischemia which can be estimated by transcutaneous oximetry [19,28]. This point is important because evidence of ischemia associated with symptoms is required to define the arterial origin of TOS [19].

In the future, it is likely that A-PPG will not replace other tests that can be used on patients with suspected TOS. Specifically, US shall remain a first-line investigation. Nevertheless, many discrepancies have been found between different technical approaches in patients with suspected TOS [29]. Therefore, we believe that the ability of A-PPG during a “Ca+Pra” maneuver to provide objective measurable, standardized, and observer-independent recording of inflow impairment could add to the holistic approach of TOS diagnosis. Whether or not it improves the diagnostic algorithm of patients with suspected TOS will need to be tested in the future.

## Figures and Tables

**Figure 1 diagnostics-11-01128-f001:**
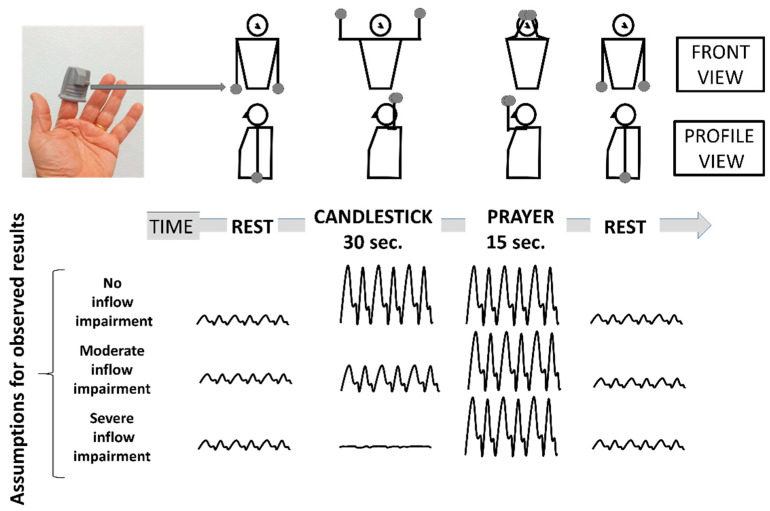
Schematic representation of the Candlestick-Prayer (“Ca+Pra”) maneuver and of expected changes in arterial digital pulse-plethysmography (A-PPG) pulse amplitude from the SpO2 soft clip (upper left image) during the different phases. With arm elevation, A-PPG pulse amplitude is expected to increase as a function of the change in hand elevation relative to the heart level. Therefore, pulse amplitude is expected to be the same in the candlestick and prayer position in normal subjects. A decrease in pulse amplitude with arm elevation is assumed to result from a severe arterial attitudinal compression (occlusion or sub-occlusion). Finally, in the case of partial (mild to moderate) compression, the amplitude observed during the candlestick position is expected to remain lower than that observed in the prayer position.

**Figure 2 diagnostics-11-01128-f002:**
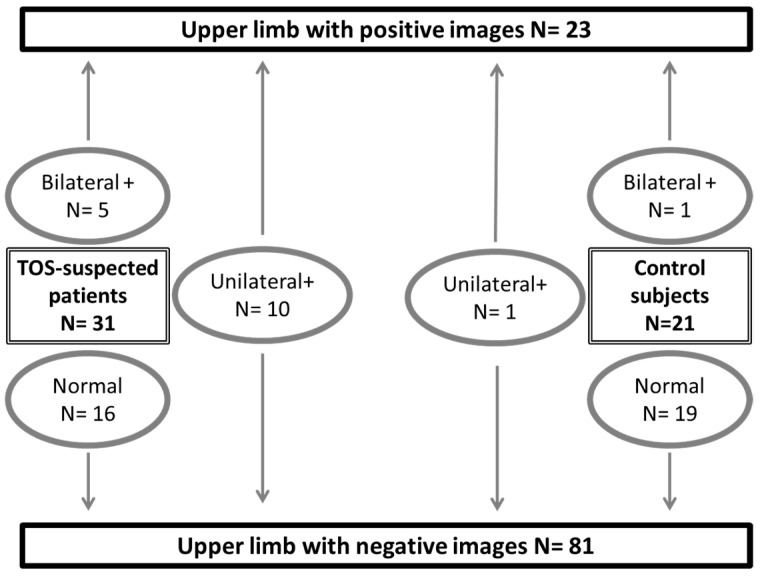
Distribution of studied subjects and results from ultrasound (+/- angiography in patients with suspected thoracic outlet syndrome: TOS) imaging. A positive result (+) is the presence of arterial compression on either ultrasound or angiography. “N” is number of observations.

**Figure 3 diagnostics-11-01128-f003:**
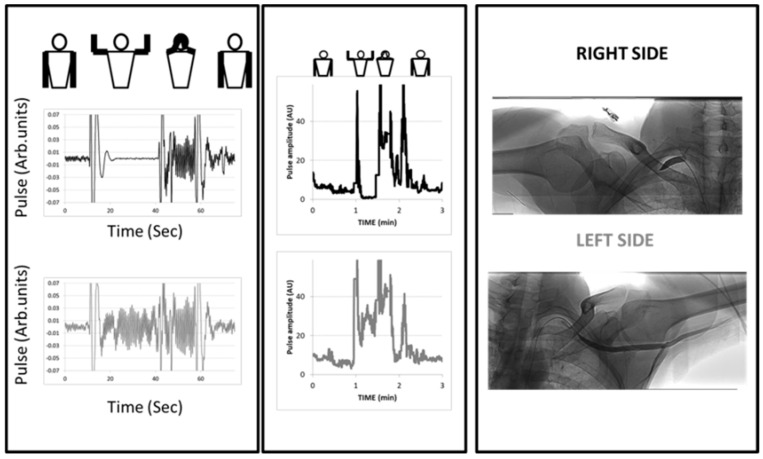
Example of an arterial digital pulse plethysmography (A-PPG) recording during the candlestick-prayer maneuver with unilateral pain on the right side and fatigability at arm elevation. Left panels represent a focus starting 10 s preceding the start of the maneuver and ending 10 s after return to the resting position on the right (black line) and left (grey line) sides. As shown, moving the arm results in ample signal artifacts. Here, pulsatility was lost 10 s after arm elevation and restored with increased amplitude during the prayer phase on the right arm, while the response on the left side was an increase from baseline during both candlestick and prayer attitude. The middle panel is the analysis of pulse amplitude throughout the period of recording (AU: from before normalization to the amplitude observed during the prayer phase). In the right panel are the angiographic results for the patient.

**Figure 4 diagnostics-11-01128-f004:**
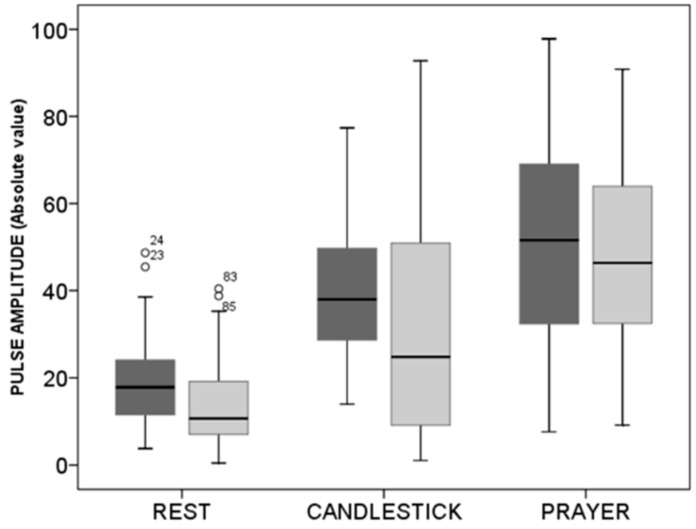
Box plots of values in arbitrary units observed at rest during the candlestick (“Ca”) and prayer (“Pra”) phases of the “Ca+Pra” maneuver in patients with suspected TOS (light grey) and in the healthy subjects (dark grey). No difference between groups was observed, but changes were significant between each position within each group. Circles are outlier values.

**Figure 5 diagnostics-11-01128-f005:**
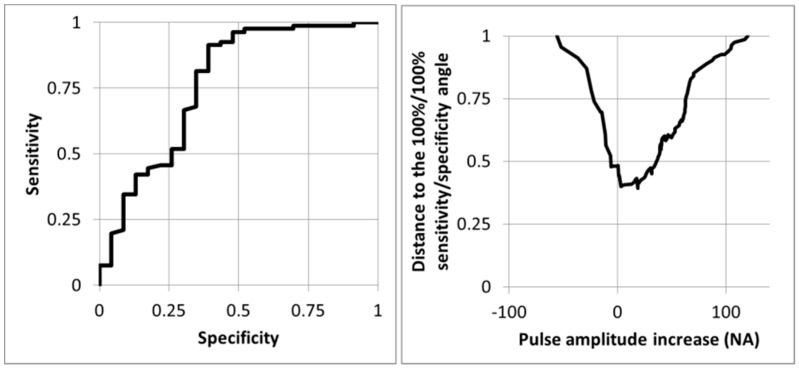
Receiver operating characteristics curve (left panel) and distance from the 100%/100% sensitivity/specificity angle (right panel) for the prediction of the presence of an arterial compression at imaging (ultrasound +/− angiography) for the 104 upper limbs using the change in pulse amplitude whereby the 100 normalized amplitude (NA) is the amplitude observed during the prayer phase of the candlestick-prayer maneuver. As shown, the optimal cut-off point is a pulse amplitude increase of 3 NA.

**Table 1 diagnostics-11-01128-t001:** Area under the receiver operating characteristics (ROC) curve (AUC) to determine the presence of a compression at imaging. AUC values are not significantly different between the three methods.

	Area under ROC Curve	Significance of the AUC	Asymptotic 95%CI Lower Limit	Asymptotic 95%CI Upper Limit
Absolute change from rest (AU)	0.715 ± 0.066	0.002	0.585	0.845
Percentage change from rest (%)	0.733 ± 0.071	0.001	0.616	0.851
Pulse amplitude change (NA)	0.765 ± 0.065	0.000	0.637	0.891

## Data Availability

The data presented in this study are available on request from the corresponding author. The data are not publicly available due to confidential medical information in the database source secured by a hospital database system.

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
