# Peer review of "Arterial Digital Pulse Photoplethysmography in Patients with Suspected Thoracic Outlet Syndrome: A Study of the “Ca+Pra” Maneuver"

_diagnostics, 2021, doi:10.3390/diagnostics11061128_

Round 1

Reviewer 1 Report

The authors have now robustly defended the original criticisms and have made relevant alterations to the paper

Reviewer 2 Report

No suggestions or comments to give after revision.

This manuscript is a resubmission of an earlier submission. The following is a list of the peer review reports and author responses from that submission.

Round 1

Reviewer 1 Report

This is a relatively novel paper investigating physiological changes in terms of arterial digital pulse plethysmography with alteration of to arm positioning during provocation manoeuvres in the investigation of TOS.

A major criticism of this study is the introduction of novel terms ("candlestick" and "prayer" positions), to describe a standard provocation manoeuvre, the Extended Arm Stretch Test (originally described by David Roos in the 1960's), at a time when there is considerable international effort being expended to standardise reporting terms and investigations in the field of Thoracic Outlet Syndrome - (reference 18, quoted by the authors: Reporting standards of the Society for Vascular Surgery for Thoracic Outlet Syndrome, J Vasc Surg,64 2016,64 e23-35). The terms "Candlestick" and "Prayer" positions do not conform to a proposed unified nomenclature, and instead impose new terminology  to describe a physiological phenomenon, which complicates standard descriptions and may confuse readers.

In their own right the authors' findings are novel and interesting with respect to the physiological phenomenon described, but as the authors acknowledge, are unlikely to influence clinical practice, because of the lack of ability to standardise the manoeuvres and because of potential artefacts in the investigations.

Duplex ultrasound remains commonplace, is used routinely, is reliable in the hands of trained operators, and remains the gold standard for non-invasive investigation of arterial compression during provocation manoeuvres. The authors criticisms that it is "hand-held" and can only be performed on one side at a time are not significant justification for alternative adoption of A-PPG.

The authors do not mention VTOS anywhere in their paper, which forms part of the spectrum of TOS disease and which is also delineated by the use of duplex ultrasound, and commonly occurs due to costoclavicular impingement. Moreover, there is no acknowledgement that some ATOS may occur due to extrinsic compression of the subclavian artery by other anomalies of scalene muscle hypertrophy or insertion, or due to the presence of abnormal bands or cervical ribs, which may not be affected by the Prayer position.

As the authors acknowledge, there is a normal distribution of arterial compression in asymptomatic individuals, and pure ATOS remains the least common presentation of the variants of TOS (NTOS, VTOS, ATOS). This calls into question the relevance of the physiological findings described by the authors across the spectrum of clinical presentations.

The authors acknowledge that there were no significant differences in absolute values in subjects with suspected TOS than in the healthy subjects both at rest and during the candlestick maneuver, and no difference was observed between the two groups  in values found at the “Pra” phase of the test. 

The patients and controls seem to have different mean baseline arm systolic pressures (Patients approx 130; controls approx 114). Would this have had any relevance to the A-PPG findings? 

The paper is well-written, the methods appropriate and clearly described, and the bibliography is up-to-date. The paper is interesting in terms of physiological observation, but this reviewer feels that the results are unlikely to have a significant impact in terms of investigation or management of ATOS.

Author Response

REVIEWER 1

GENERAL RATING

 (x) English language and style are fine/minor spell check required

We apologize if the revision has introduced additional grammar style or spelling errors and we are ready to correct them wherever necessary.

Yes

Can be improved

Does the introduction provide sufficient background and include all relevant references?

( )

(x)

Is the research design appropriate?

(x)

( )

Are the methods adequately described?

(x)

( )

Are the results clearly presented?

(x)

( )

Are the conclusions supported by the results?

( )

(x)

Comments and Suggestions for Authors

This is a relatively novel paper investigating physiological changes in terms of arterial digital pulse plethysmography with alteration of to arm positioning during provocation manoeuvres in the investigation of TOS.

A major criticism of this study is the introduction of novel terms ("candlestick" and "prayer" positions), to describe a standard provocation manoeuvre, the Extended Arm Stretch Test (originally described by David Roos in the 1960's), at a time when there is considerable international effort being expended to standardise reporting terms and investigations in the field of Thoracic Outlet Syndrome - (reference 18, quoted by the authors: Reporting standards of the Society for Vascular Surgery for Thoracic Outlet Syndrome, J Vasc Surg,64 2016,64 e23-35). The terms "Candlestick" and "Prayer" positions do not conform to a proposed unified nomenclature, and instead impose new terminology to describe a physiological phenomenon, which complicates standard descriptions and may confuse readers.

We agree with the reviewer that the terminology is confusing and that there are already multiple expressions that have been proposed to describe the maneuvers performed to investigate patients with TOS, the Roos Test, the Adson's Test, the Scalene test, the Halsted's test, the Wright's test, the Eden test, the “Military brace test”, the Cyriax release test, the Upper limb tension test (ULTT), extended/elevated arm stress test (EAST), etc… with some of them referring to the same or almost the same procedure.

We are perfectly aware of the JVS2016 publication and agree to the recommendation that the “Roos test” expression should be used (whether or not it includes hand movements), and that it should replace the “Extended Arm Stress Test” (EAST) sometimes also described as the “Elevated arms stress test”, sometimes described also as the 90° AER test (abduction external rotation :Sanders et al JVS 2007).

Nevertheless, we do not share the idea, and are not keen to follow the suggestion, of the reviewer to remove “Ca” and “Pra” positions as abbreviation of the positions used for our procedure for the reasons explained below:

On the one hand, to the best of our knowledge nobody ever suggested to move the elbows forward following the elevation of the arm in 90° abduction and external rotation. Then it seems to us a bit unfair to suggest that both “The terms "Candlestick" and "Prayer" positions do not conform to a proposed unified nomenclature. How could we conform to such a nomenclature if the position (at least for the prayer position with arm elevated forward) has never been proposed? We just and only suggested that this position was mimicking a praying person.

On the other hand, we totally agree with the reviewer that we should not have referred to a “candlestick test” but only to a candlestick position. The surrender / candlestick expressions (Candlestick is largely used in continental Europe) are illustrations of body position itself and should not be used to describe the whole Roos test. We completely agree that the use of the expression “candlestick test” a few times in the manuscript was absolutely inappropriate, and was adding to the confusion. We apologize that it occurred, and these errors have be removed throughout. Candlestick and prayer are only illustrations of the positions and the complete procedure that we used was a combination of these two positions.

Finally, whether or not the complete ”Ca+Pra” maneuver should be considered a “rerevised Roos test”, a “PPG-adapted Roos test” a “modified Ross test” or a “Roos+Prayer test” is possible but clearly the complete “90°-abduction-rotation-followed-by-elevation-in-front-of-the-body-followed-by-arm-lowering” procedure is not a Ross test per se…because nobody ever suggested that after the position used for the Roos test, arm should be positioned in front of the subjects before they are lowered.

In their own right the authors' findings are novel and interesting with respect to the physiological phenomenon described, but as the authors acknowledge, are unlikely to influence clinical practice, because of the lack of ability to standardise the manoeuvres and because of potential artefacts in the investigations.

We share the reviewer view that the most immediate applicability of our finding is physiopathology, and we hope that the reviewer will share our view that generally progresses in physiopathology knowledge are quite likely to result in future improvements in patient care (among which diagnosis is only one aspect). We disagree with the idea that A-PPG during the Ca+Pra maneuver would result in a lack of standardization. A-PPG during the Roos test cannot be standardized… thus the debate on what should be considered an abnormal response (Geven, Gegoudis, Admas limits?). Clearly the Prayer position is there to standardize the results observed during the surrender/Candlestick position. One could claim that an elevation before or following the end of a Roos test would reach post-test maximal amplification and would allow similar standardization. This is true but would introduce the question of how long should one wait between the two recordings? 

Duplex ultrasound remains commonplace, is used routinely, is reliable in the hands of trained operators, and remains the gold standard for non-invasive investigation of arterial compression during provocation manoeuvres. The authors criticisms that it is "hand-held" and can only be performed on one side at a time are not significant justification for alternative adoption of A-PPG.

We do apologize if the manuscript was unclear and if the first impression of the reviewer was that we proposed A-PPG as an alternative technique to replace ultrasounds. This is clearly not our view, as underlined in our conclusion (and perspectives) paragraph. All patients in our department have duplex (triplex in fact) ultrasound investigations and we are quite happy with it. We are not advocating for the replacement of ultrasound by PPG. Clearly to our view PPG is an additional approach.

The reviewer suggests that the arguments provided in the manuscript to underline US limitation are insufficient. The first arguments provided in the manuscript is the hand-held character. We agree that it is generally not a major issue except if a 1.60 m tall operator has to deal with a 1.95m tall patient and tests are performed standing. Nevertheless, then the test can be performed in the sitting position. It can also be an issue because it is sometimes difficult to follow patient’s movements without losing the image or signal of interest. Last, it is also an issue if one aims at monitoring and recording patient’s movements because of the presence of the ultrasonologist near the patient, as will be explained below. The second argument in the manuscript is that only one side can be studied at a time… Contrary to the reviewer, we believe that it is an issue because we frequently observed compensatory attitudes with patients lowering the contralateral shoulder when performing single-arm abduction. This results in the true abduction angle being smaller than the expected 90°. There, one should argue that on the one hand the observer just has to take attention to it or to record the patient’s position. The former argument is true although focusing on contralateral arm and shoulder position is not so easy because the operator already focuses on US images, the latter argument suggesting direct recording of arm position is a very important point and will be discussed below. One should further argue that the operator can ask the patient to elevate the two arms simultaneously even if one arm is not recorded and that the other arm will be done the same way. In our experience it is often an issue in patients that have quite severe positional symptoms, and many patients report pain and fatigability when tests are repeated multiple time or when tests are prolonged. Recording the two sides simultaneously would decrease the duration/repetition of test and might lead to decreasing patient’s discomfort.

Further, as rightly suggested by the reviewer, US requires “Trained operator”. It is clearly not an issue if investigation is performed in referral centers but becomes an issue in the perspective of primary care and becomes an issue in the perspective of potential epidemiological study or multicenter studies. If one aims at performing large scale multicenter studies, an observer independent approach is preferable to reduce the risk of between-center differences and sensitivity to observer’s experience. This training requirement issue has been added to the manuscript to better argue for the interest of studying other techniques than US.

Last, there is in fact another reason why we aimed to focus on APPG, beyond those that are now appearing in the manuscript. Nevertheless, this one will not appear in the manuscript because of confidentiality issues. Reliability is an issue in TOS positional investigations. We observed that test-retest reliability of the vascular response during provocative tests is not as good as generally assumed. Clearly lack of reliability may result of insufficient expertise. Nevertheless reason for this variability also relies in our opinion on small changes that occur in arm and forearm position from one test to another (or during the test itself with the patients insensibly lowering the arm due to pain or fatigue). As a result of this assumption, our aim is to develop a device integrating a continuous recording of upper limb position (through a Kinect camera) and a continuous recording of the vascular response (PPG) because PPG is not requiring the presence of an operator next to the patient (that would interfere with the Kinect analysis). Such a device would estimate the maneuver really performed by the patient and to our opinion could improve the methods and facilitate the interpretations of the results. It could also facilitate the recordable determination of the abduction angle that induces compression of the neurovascular bundle which might become in the future an index to discriminate “physiological” versus “pathological” positional compressions (We hypothesize that a compression yet occurring at 40° abduction will become symptomatic while one occurring only at 80° abduction will remain asymptomatic and should be considered physiological). Simultaneously recording arm position and the vascular response, is also of particular interest if investigations are performed by technicians without the presence of the physician, to allow post-test quality control.

The authors do not mention VTOS anywhere in their paper, which forms part of the spectrum of TOS disease and which is also delineated by the use of duplex ultrasound, and commonly occurs due to costo-clavicular impingement. Moreover, there is no acknowledgement that some ATOS may occur due to extrinsic compression of the subclavian artery by other anomalies of scalene muscle hypertrophy or insertion, or due to the presence of abnormal bands or cervical ribs, which may not be affected by the Prayer position.

This is an important and interesting point with in fact two different aspects in the remark.

First, the reviewer is perfectly right that we did not mention VTOS in the manuscript. In fact, neither did we mention ATOS or NTOS. We were very careful not to classify our patients into NTOS, VTOS or NTOS sub-groups at all, because it is not based on the sole results of investigations but needs integration of symptoms and signs (some of which are not so easily or uniformly reported by the patients or not easy to analyze (i.e. paresthesia of the whole hand versus paresthesia of the 4th and 5th finger), or may depend on the level of the compression. The second point here is that even in NTOS, the presence of an arterial compression (either determined by pulse palpation stethoscope auscultation, Doppler recording, PPG recording, saturometry, etc…) during provocative maneuvers is widely used to provide arguments for positional compression of the neurovascular bundle. The last point is that, to our opinion, A-PPG is probably not the optimal tool to detect the permanent (Paget Schroetter syndrome) or transient (Mc Cleery syndrome) impaired outflow. Low pass filtering of PPG signal would clearly be more adapted but it is another story.

Second, it is perfectly right that extrinsic compressions occurring at different levels (and not the costoclavicular angle) may not be affected by the Prayer position. This was not discussed in the manuscript and was clearly missing. This important point has been added in the limitation paragraph, the following way “compression of the neural or vascular structures may occur at various levels as they pass through narrow passageways to the arm, including the inter-scalene triangle, the costo-clavicular space, the sub-coracoid space (pectoralis minor compression), or in front of the humeral head. Compression may also result from the presence of abnormal bands, tumors, cervical ribs, bone prominence following healing of fractures. These different entities may not be affected the same way by the Prayer position. Future studies are required to analyze the response to the Ca+Pra maneuver in perspective of the level(s) and nature of arterial compression(s).” We thank the reviewer for this important and clever suggestion.

As the authors acknowledge, there is a normal distribution of arterial compression in asymptomatic individuals, and pure ATOS remains the least common presentation of the variants of TOS (NTOS, VTOS, ATOS). This calls into question the relevance of the physiological findings described by the authors across the spectrum of clinical presentations.

We thank the reviewer for this comment to which we have already largely answered above when it comes to discuss why we did not mention VTOS (nor actually NTOS or ATOS) in the present manuscript. We advocate that it would not be true to suggest that arterial investigations are only performed in patients presenting symptoms of arterial-TOS. Clearly, at least to the best of our knowledge about usual practice of most of our colleagues, detecting (vascular) signs of positional compression of the neurovascular bundle, regardless of how TOS will finally be classified into one of the three sub-group, is a diagnostic routine. Most physicians rely on Doppler and or ultrasound imaging, some rely on auscultation of the bruit appearing with arterial stenosis. Some physicians use PPG or pulse oximetry to detect the arterial compression, Some physicians require angiography (Arteriography, Angiosan or Angio MRI) others do not. Even transcutaneous oxygen pressure was proposed recently to detect the ischemia resulting from arterial occlusion. Last, due to the provocative maneuvers being sometimes non reproducible, we observed that many physicians expect reasonable consistent results to give a formal answer. Overall, none of the above tools during positional maneuvers solely qualifies patients to be in one of the TOS sub-groups, they only prove that a positional compression is present. Similarly, none can confirm that symptoms are related to the compression because compressions can be observed in a large amount of asymptomatic patients and other diseases can be responsible for symptoms. Why should the A-PPG results be applicable to ATOS only (the rarest form of TOS) while results of other techniques are used to define the presence of compression regardless of the sub-group classification?

The authors acknowledge that there were no significant differences in absolute values in subjects with suspected TOS than in the healthy subjects both at rest and during the candlestick maneuver, and no difference was observed between the two groups in values found at the “Pra” phase of the test.  The patients and controls seem to have different mean baseline arm systolic pressures (Patients approx 130; controls approx 114). Would this have had any relevance to the A-PPG findings? 

Thank you for this question. Plethysmographic indices and arterial waveform-derived indices are physiologically related. It cannot be excluded that the higher systolic pressure observed in patients as compared to controls might be one of the factors that resulted in the slightly higher (although not significant) pulse amplitudes found in patients compared to controls. Many studies showed a relationship between changes in pulse amplitude and changes in blood pulse pressure, among many others Cannesson M et al.. Critical Care 2005;9:R562–8, Sandroni, C et al. Intensive Care Medicine volume 38, pages1429–1437(2012), Golparvar, M. et al. Anesthesia & Analgesia: December 2002 - Volume 95 - Issue 6 - p 1686-1690. Nevertheless, if monitoring of arterial pressure or pulse pressure changes accurately correlates to monitoring of pulse amplitude  changes, when it comes to absolute values the relationship is poor, probably because of the semi-quantitative character of PPG. Since no difference was observed on PPG pulse amplitude between our two groups we did not comment on this question and do not think that this should add to the discussion. Nevertheless, if the reviewer disagrees with our position a short paragraph could be added in the discussion.

The paper is well-written, the methods appropriate and clearly described, and the bibliography is up-to-date. The paper is interesting in terms of physiological observation, but this reviewer feels that the results are unlikely to have a significant impact in terms of investigation or management of ATOS.

We thank the reviewer for underlying the interest of the present approach on a physiological point of view. As previously underlined, we share the reviewer view that the most immediate applicability of our finding is physiopathology, but we hope that the reviewer will share our view that generally progresses in physiopathology knowledge are quite likely to result in future improvements in patient care (among which diagnosis is only one aspect). Further and once again we disagree that the present finding would be of potential interest only for ATOS (the rarest form of TOS) while results of other techniques are used to define the presence of a positional compression regardless of the sub-group classification?

Reviewer 2 Report

The authors examined whether in patients diagnosed with the thoracic outlet syndrome (TOS) eventual compression that occurred in the candlestick position (“Ca”) could be released by moving the arms forward, mimicking a prayer position (“Pra”), thus facilitating arterial digital pulse plethysmography (A-PPG) results interpretation. In this regard they determined the optimal pulse amplitude (PA) change from rest to predict compression at imaging with receiver operating characteristics  (ROC). They observed that A-PPG during a “Ca+Pra” test provided demonstrable proof of inflow impairment and increased the sensitivity of A-PPG for the detection of arterial compression as determined by imaging and concluded that the absence of an increase in PA during the “Ca” phase of the “Ca+Pra” maneuver  should be considered indicative of arterial inflow impairment.

Comments /queries

1.The authors state : “Using the results of ultrasound and angiography to characterize the presence of a compression with ROC analysis, the larger area was observed for the change from rest to the “Ca” position normalized to the value found during the “Pra” phase of the test and expressed in NA units, as shown in table 1.“ Which method was used to compare ROC curves? Please present the results supporting this conclusion.

2.In the last paragraph the authors state: “ In the future, it is unlikely that A-PPG will replace other tests that can be used on patients with suspected TOS. Nevertheless, many discrepancies have been found between different technical approaches in patients with suspected TOS [28]. Therefore, we believe that the ability of A-PPG during a “Ca+Pra” maneuver to provide objective measurable, standardized and observer-independent recording of inflow impairment could add to the holistic approach of TOS diagnosis. Whether or not it improves the diagnostic algorithm of patients with suspected TOS will need to be tested in the future“. I really do not understand the meaning of this paragraph. What was the purpose of this study?

Author Response

REVIEWER 2

Open Review

English language and style

( ) Extensive editing of English language and style required
(x) Moderate English changes required
( ) English language and style are fine/minor spell check required
( ) I don't feel qualified to judge about the English language and style

We do apologize that despite reviewing and editing by an expert company to which we requested an editing in American English, language and style are still imperfect. Could it be due to some typo errors than passed under their radar? Could it be resulting from differences between British English and American English? Could it be due to the use of some unsatisfactory medical expressions? Could it be due to minor changes that we did after this editing?  Whatever, we would be more than happy to improve this point under the present reviewer suggestions.

Must be improved

Does the introduction provide sufficient background and include all relevant references?

(x)

Is the research design appropriate?

(x)

Are the methods adequately described?

(x)

Are the results clearly presented?

(x)

Are the conclusions supported by the results?

(x)

We did our best to improve the manuscript according to the comments of the reviewer and hope that our improvements are fulfilling the reviewer expectations.

Comments and Suggestions for Authors

The authors examined whether in patients diagnosed with the thoracic outlet syndrome (TOS) eventual compression that occurred in the candlestick position (“Ca”) could be released by moving the arms forward, mimicking a prayer position (“Pra”), thus facilitating arterial digital pulse plethysmography (A-PPG) results interpretation. In this regard they determined the optimal pulse amplitude (PA) change from rest to predict compression at imaging with receiver operating characteristics  (ROC). They observed that A-PPG during a “Ca+Pra” test provided demonstrable proof of inflow impairment and increased the sensitivity of A-PPG for the detection of arterial compression as determined by imaging and concluded that the absence of an increase in PA during the “Ca” phase of the “Ca+Pra” maneuver  should be considered indicative of arterial inflow impairment.

Thank you for this excellent synthesis of our work.

Comments /queries

1.The authors state : “Using the results of ultrasound and angiography to characterize the presence of a compression with ROC analysis, the larger area was observed for the change from rest to the “Ca” position normalized to the value found during the “Pra” phase of the test and expressed in NA units, as shown in table 1.“ Which method was used to compare ROC curves? Please present the results supporting this conclusion.

We apologize that the method used was not presented in the method section. The reference to the method used by our biostatistician (P Saulnier) has been added to the reference list. We have. Area of normalized changes was significant from values at rest but resting values are not to be used as diagnostic tests and these areas have been removed from table 1. We now keep only changes in absolute values, percent changes or changes in, normalized values. (Absence of) significance is presented in the legend of the table and a short paragraph has been added to the discussion.

2.In the last paragraph the authors state: “ In the future, it is unlikely that A-PPG will replace other tests that can be used on patients with suspected TOS. Nevertheless, many discrepancies have been found between different technical approaches in patients with suspected TOS [28]. Therefore, we believe that the ability of A-PPG during a “Ca+Pra” maneuver to provide objective measurable, standardized and observer-independent recording of inflow impairment could add to the holistic approach of TOS diagnosis. Whether or not it improves the diagnostic algorithm of patients with suspected TOS will need to be tested in the future“. I really do not understand the meaning of this paragraph. What was the purpose of this study?

Thank you for these comments. May be we should have titled the last part (part 5) “Conclusions and perspectives” instead of “conclusions” only, but this would not fit the journal recommendations. Probably, the paragraph was not sufficiently clear. We are convinced that progresses can be made in the understanding of TOS (physiopathology) and managements (clinical medicine) of patients referred for TOS. Obviously the present study is insufficient to provide arguments for the routine use of the present approach and future experiments, external validation, evaluation in a large group, cost-effectiveness of the tool, etc… have to be evaluated and studied. The present work is only a step forward. TOS remains a difficult diagnosis based on a holistic approach including symptoms, clinical investigations, functional tests, ultrasound and radiological imaging and physician face (at least in our experience) numerous cases of complex interpretation of the available evidences … (not forgetting the possibility of co-morbid conditions that may interfere with the final diagnosis). Last, we believe that the major reason why no consensus exists about what should be considered an abnormal A-PPG response during the Roos test, is because: first A-PPG is a semi-quantitative tool, and second the individual response resulting from arm elevation but without 90° abduction and external rotation, is unknow. Accounting for the reviewer comment, the above sentence has been added to the introduction to try to have readers better understand the interest of the present work.

Reviewer 3 Report

Your work is interesting and tries to find a new method of validating a diagnosis that is difficult to make. I would have liked you to be firmer and less modest in your discussions and conclusions, given the effort made. Nice job!

Author Response

REVIEWER  3

Open Review

English language and style

( ) Extensive editing of English language and style required
( ) Moderate English changes required
( ) English language and style are fine/minor spell check required
(x) I don't feel qualified to judge about the English language and style

Yes

Does the introduction provide sufficient background and include all relevant references?

(x)

Is the research design appropriate?

(x)

Are the methods adequately described?

(x)

Are the results clearly presented?

(x)

Are the conclusions supported by the results?

(x)

Comments and Suggestions for Authors

Your work is interesting and tries to find a new method of validating a diagnosis that is difficult to make. I would have liked you to be firmer and less modest in your discussions and conclusions, given the effort made. Nice job!

Thank you. We really appreciate that you share our enthusiasm but we have to be careful with some readers that might not. 

Round 2

Reviewer 1 Report

The authors have made a significant and robust attempt to defend their position with respect to the reviewers' criticisms, and to make reasonable adjustments. On that basis, my feeling is that the paper is fit for publication on a medium priority basis. 

Reviewer 2 Report

Major Comment

206 Imaging (ultrasound or angiography) showed the presence of a compression in 20 of

207 the 62 upper limbs studied in patients with suspected TOS but in only 3 of the 42 upper

208 limbs in the healthy subjects. Therefore, the prevalence of positive imaging was 22.1%

209 among the 104 studied upper limbs. Using the results of ultrasound and angiography to

210 characterize the presence of a compression with ROC analysis, the larger area was ob-

211 served for the change from rest to the “Ca” position normalized to the value found dur-

212 ing the “Pra” phase of the test and expressed in NA units, as shown in table 1.

213 Table 1: Area under the receiver operating characteristics (ROC) curve (AUC) to determine the

214 presence of a compression at imaging. AUC are not significantly different between the three

215 methods.

It is obvious that Table 1, which has been radically changed  compared with the previous version, is discordant with the text which  has virtually remained the same!

Minor comment

AUC takes values from 0 to 1, where a value of 0 indicates a perfectly inaccurate test and a value of 1 reflects a perfectly accurate test. I have never seen values as those of Table 1.